# Advances in Mass Rearing *Pseudophilothrips ichini* (Hood) (Thysanoptera: Phlaeothripidae), a Biological Control Agent for Brazilian Peppertree in Florida

**DOI:** 10.3390/insects12090790

**Published:** 2021-09-03

**Authors:** Dale A. Halbritter, Min B. Rayamajhi, Gregory S. Wheeler, Jorge G. Leidi, Jenna R. Owens, Carly A. Cogan

**Affiliations:** Invasive Plant Research Laboratory, USDA, Agricultural Research Service, 3225 College Ave., Ft. Lauderdale, FL 33314, USA; min.rayamajhi@usda.gov (M.B.R.); greg.wheeler@usda.gov (G.S.W.); jorge.leidi@usda.gov (J.G.L.); jenna.owens@usda.gov (J.R.O.); carly.cogan@usda.gov (C.A.C.)

**Keywords:** aggregation, passive trapping, phytophagous thrips, *Schinus terebinthifolia*

## Abstract

**Simple Summary:**

*Pseudophilothrips ichini* is a recently approved biological control agent to control the highly invasive Brazilian peppertree in Florida, USA. Methods for producing large numbers of these thrips are needed to ensure enough are available for field release across the state. Prior to approval for field release in 2019, thrips colonies were kept in small cylindrical cages that fit in limited quarantine space. We developed novel techniques to expand from small colony maintenance to large-scale production. We first quantified the productivity of small cylinders, which each produced an average of 368 thrips per generation. Given the amount of maintenance the cylinders required, we investigated larger cages to see if greater numbers of thrips could be produced with less effort. Larger acrylic boxes produced an average of 679 thrips per generation. The final advancement was walk-in screen cages that each produced an average of 13,864 thrips per generation. Screen cages produced up to 37 times more thrips per enclosure while requiring significantly fewer personnel hours. The large screen cages efficiently produced thousands of thrips weekly, permitting us to sustain mass distribution in the field. The advances made here contribute to the published methods on thrips rearing and are among the few that focus on mass rearing thrips as beneficial insects.

**Abstract:**

*Pseudophilothrips ichini* is a recently approved biological control agent for the highly invasive Brazilian peppertree in Florida, USA. Prior to approval for field release in 2019, thrips colonies used for host specificity testing were produced and maintained in small cylinders to fit in restricted quarantine spaces. This next segment in the classical biological control pipeline is mass production and distribution of *P. ichini*. To accomplish this, we developed novel techniques to expand from small colony maintenance to large-scale production. We first quantified the productivity of the small cylinders, each containing a 3.8 L potted plant and producing an average of 368 thrips per generation. Given the amount of maintenance the cylinders required, we investigated larger cages to see if greater numbers of thrips could be produced with less effort. Acrylic boxes (81.5 × 39.5 × 39.5 cm) each contained two 3.8 L plants and produced an average of 679 thrips per generation. The final advancement was large, thrips-proof Lumite^®^ screen cages (1.8 × 1.8 × 1.8 m) that each held six plants in 11.4 L pots and produced 13,864 thrips in as little as 5 wk. Screen cages and cylinders had the greatest thrips fold production, but screen cages required ten times less labor per thrips compared to either cylinders or boxes. The efficiency of these large screen cages ensured sustained mass production and field release capacity in *Schinus*-infested landscapes. The screen cage method is adapted and used by collaborators, and this will expand the literature on beneficial thrips mass rearing methods.

## 1. Introduction

Brazilian peppertree, *Schinus terebinthifolia* Raddi (Sapindales: Anacardiaceae), a woody shrub, often growing as a multi-stemmed small tree, is native to Argentina, Brazil, Paraguay, and Uruguay [1], (Wheeler, unpublished data). The plant has become an invasive weed in Florida [2,3], California [4], Hawaii [5], South Texas [6], and Australia [7,8]. In Florida, Brazilian peppertree (*Schinus*, henceforth) invades a wide variety of urban and natural landscapes, the latter including cypress plant communities [9], hammocks [10], pinelands [11], and mangrove forests [12]. Mechanical and chemical control methods are costly [13], and repeated applications are needed to keep the weed under control. Consequently, efforts to develop a biological control program began with surveys for agents in Brazil and Argentina in the 1950s and 1960s [14,15], with the most recent surveys starting in 2005 [16]. *Pseudophilothrips ichini* (Hood) (Thysanoptera: Phlaeothripidae) was a promising candidate for Florida populations of *Schinus* [16,17] and was approved by USDA-APHIS for field releases in Florida in May 2019 [18].

*Pseudophilothrips ichini* (thrips, henceforth) feeds gregariously as larvae and adults on newly flushing stems and leaves (Figure 1). We will refer to feeding immatures as larvae, as in Hoddle et al. [19]. The thrips feed by piercing the cell walls and drinking plant fluids, and high thrips densities result in the death of the new leaves and succulent apical meristems. The life cycle of the thrips is comprised of the egg, two larval instars, three pupal instars, and the adult. Eggs are 0.4 mm, first instar larvae are 0.7 mm, second instar larvae are 1.0 mm, pupae range from 1.6 to 2.1 mm, and adults are 2.5 mm in length (see Figure 2 in [20]). Eggs are laid topically in tight spaces such as leaf axils, rachis depressions, and between veins of new leaflets near meristems. Larvae tend to remain on the stem they were oviposited on but will wander the plant in search of new stem tips if needed to complete their growth to maturity. The bright orange second instar larvae either drop off the plant or walk down to the soil to pupate in the duff. The black, winged adults are weak fliers and tend to form nearby aggregations shortly after emergence. Complete development from egg to adult varies with temperature, taking from 34.5 d at 20 °C to 18.9 d at 30 °C at constant temperatures [21].

A dependable and efficient method for mass producing *P. ichini* is critical to ensure that large numbers of healthy thrips are available for field releases to establish sustaining populations on *Schinus* infestations ranging from Key West in the south to northcentral Florida, and potentially in other regions of the southern United States. Most of the literature on rearing thrips addresses polyphagous crop and horticultural pests produced for experimental studies [22,23] or rearing predatory thrips for augmentative biological control [24]. While some techniques such as sanitation and microclimate regulation can be adapted, species-specific dietary needs and developmental biology need to be objectively studied and optimized for adaption in mass rearing. With respect to *Pseudophilothrips* spp., a technique using 45 cm tall by 15.5 cm diameter clear, vented acrylic cylinders fit over *Schinus* plants growing in 3.8 L pots has been successfully used to rear and maintain small colonies for experimentation and host range testing in quarantine facilities for all life stages [17,20,25] or early life stages of the thrips [21,26]. Cuda et al. [26] report a combined monthly total of up to 3153 thrips (mixture of *P. ichini and P. gandolfoi*) produced using cylinders for oviposition and initial larval development. Details on cylinder productivity in other studies pertain to thrips production for specific experimental purposes such as host range testing [17,20,21,25,26], rather than large-scale colony output for field releases.

With the shift from maintaining colonies in limited quarantine space to mass production to meet the demand for field release, we explored increasing cage sizes to progress toward large-scale production. Rearing of *Boreioglycaspis melaleucae* Moore (Hemiptera: Psyllidae), a sap-feeding agent for *Melaleuca quinquenervia* S.T. Blake (Myrtaceae), similar in size and feeding behavior to *P. ichini*, was accomplished, with whole potted plants in mesh bags [27] or in screen cages [28]. Large 1.8 × 1.8 × 1.8 m screen cages worked well for *B. melaleucae* mass production for field release, as nearly 800,000 psyllids were produced in one year (Rayamajhi, unpublished data), and this served as the basis for the large screen cage method we ultimately developed and adapted for *P. ichini*. We used various concepts of quality control in mass rearing summarized in Leppla and Ashley [29] while developing and evaluating our methods, including careful consideration of *P. ichini* biology, process control via environmental monitoring, and evaluation of discrete cohorts.

## 2. Materials and Methods

### 2.1. Plant Production

*Schinus* drupes (i.e., fruits) were collected each winter from wild plants growing in the vicinity of the Invasive Plant Research Laboratory in Davie, FL, USA. To prepare for germination as needed throughout the year, exocarps were removed and seeds were soaked in water for 24 h. Prepared seeds were sewn into germination trays with Fafard^®^ Superfine Germinating Mix (Sun Gro Horticulture Distribution, Inc., Agawam, MA, USA). The wells in the germination trays were seeded individually and trays had clear plastic covers over them until germination. Trays were placed on a bench in a rainproof screen house and irrigated with overhead sprinklers supplied with tap water for 45 min twice daily to maintain moisture in the growth medium. Once the first set of leaves fully expanded, seedlings were individually transferred to 1.5 L pots (13.5 cm tall, 12.0 cm diameter) filled with a 1:1 mixture of Sun Gro Professional Growing Mix (Sun Gro Horticulture Distribution, Inc., Agawam, MA, USA) and PRO-MIX BX Growing Medium (Premier Horticulture, Inc., Quakertown, PA, USA). Plants were allowed to grow 30–45 cm in height and then trimmed back to ca. 15 cm to stimulate lateral branching with more succulent tips with new leaves for thrips oviposition and feeding as the plants continued to mature. These potted plants were maintained in the rainproof screen house and irrigated in the same fashion as the seedlings.

Once basal stem diameters reached approximately 10–15 mm, plants were transferred into 3.8 L pots (17.5 cm tall, 16.0 cm diameter) containing the 1:1 growth medium mixture described above. The majority of the 3.8 L plants were moved to a larger, uncovered screen house and overhead sprinklers irrigated daily before dawn for 1 h with water from an adjacent retention pond. Plants of optimum quality for thrips, i.e., with 10–20 actively flushing succulent stem tips, were moved outdoors onto ground covered with a black landscaping tarp (cat # 170066; Universal Enterprises Supply Corp, Pompano Beach, FL, USA). Overhead sprinklers irrigated with tap water for 1 h daily. A portion of the plants in 3.8 L pots were transplanted to 11.4 L pots (23 cm tall, 27 cm diameter) filled with the 1:1 growth medium mixture to facilitate growth into larger plants to be used in the screen cages. The individual plants in 11.4 L pots were held at an outdoor area on the ground covered in the black landscaping tarp and were hand irrigated once or twice daily for 8 s each using well water. Routine pruning continued for all three pot sizes at all growing locations, which occurred every 4 wk for plants for 1.5 L pots, every 5 wk for 3.8 L pots, and every 16 wk for 11.4 L pots.

All plants were routinely fertilized to accelerate growth and maximize plant quality. Evidence suggests more *P. ichini* thrips can be produced on plants fertilized at intermediate levels [25], but the thrips may balance their life history strategies to compensate for differences in plant quality associated with fertilization [30]. Regardless, the benefits of rapid plant growth and the need for abundant material necessitated the use of fertilizer. We used a blended solution of 2.6 mL/L Peters Professional (20-20-20 N-P-K; Everris NA, Inc., Dublin, OH, USA) and 1.3 mL/L KeyPlex^®^ 350 (KeyPlex, Winter Park, FL, USA) as a readily available liquid fertilizer and source of secondary micronutrients, respectively. The liquid blend was applied every 2 wk via backpack sprayer onto the foliage to runoff, and the growth medium surface was dampened. This equated to approximately 2, 5, and 10 s of spray time for 1.5 L, 3.8 L, and 11.4 L plants, respectively. For a slow-release fertilizer, Osmocote^®^ Blend (18-5-12 N-P-K; Everris NA, Inc., Dublin, OH, USA) granules were added to the growth medium surface every 3 mo in amounts based on the medium label rate and adjusted for the size of each pot.

Pesticides were used to suppress or eliminate generalist herbivores, thereby maintaining optimal plant quality (i.e., undamaged new growth) and minimizing competition with *P. ichini* when plants were inoculated with thrips. We primarily relied on a weekly foliar spray with a solution of 3.9 mL/L Green Cleaner (ARBICO Organics, Oro Valley, AZ, USA) and 7.8 mL/L Neem Oil (Bonide, Oriskany, NY, USA). Each plant was sprayed such that both surfaces of the leaves were wetted to runoff. This method was largely effective against aphids, black vine thrips (*Retithrips syriacus* Mayet Thysanoptera: Thripidae), red-banded thrips (*Selenothrips rubrocinctus* (Giard) Thysanoptera: Thripidae), and broad mites (*Polyphagotarsonemus latus* (Banks) Acari: Tarsonemidae). Given the Green Cleaner/Neem Oil treatment worked as a contact pesticide and can be rinsed off with little residue remaining, plants could be readily available for rearing *P. ichini* after they dried. However, occasional outbreaks of one or more of the aforementioned pests occurred, and we used foliar spray applications of either acephate (Ortho Systemic Insect Killer^®^; Scotts Company, LLC, Marysville, OH, USA) or abamectin (Avid^®^ 0.15EC Miticide/Insecticide; Syngenta, Greensboro, NC, USA) as needed per the label rate for controlling the pest(s) in question. To allow sufficient time for the pesticides to degrade, plants treated with acephate were not used for at least 4 wk [31], and plants treated with abamectin were not used for at least 2 wk (Halbritter, unpublished data). As a final pest reduction measure before a plant was introduced to the thrips colony, its outer root ball was briefly inspected for ant nests, spiders, or other potential predators, and the aerial portion of the plant was checked for potential remnants of the aforementioned pests. Plants with ant nests were not used, otherwise, spiders and other predators were removed by hand. Plants were then inverted and dunked in a soapy water bath (0.3 mL/L of Joy^®^ dishwashing liquid; Proctor and Gamble, Cincinnati, OH, USA) for 3 min. This helped to reduce any pests that may have landed on the plants after the routine foliar sprays. Lastly, plants were rinsed with tap water and allowed to air dry before being added to the thrips colony.

### 2.2. Thrips Rearing Cages

All rearing activity occurred at the Invasive Plant Research Laboratory in Davie, FL, USA. Initially, we continued to use the 45 cm tall, 15.5 cm diameter clear, vented acrylic cylinders (Figure 2A) as thrips were gradually moved out of quarantine after the release permit was issued by APHIS. Roughly 45 cm tall, pest-free plants in 3.8 L pots were used to set up individual cylinders. Larger and older leaves that are less preferred by thrips were pruned from the plants to prevent foliage from being too densely packed when the stems were brought closer together to fit into the cylinders. Often, densely packed cylinders created environments favorable for mold and condensation on the cylinder walls. Long-fiber sphagnum moss (Natural Pack, Inc., Reading, PA, USA) was added to the growth medium surface of each pot, enough to almost reach the top of the pot. Adding moss to the cylinders has been shown to increase *P. ichini* production, as it likely provides a more suitable microhabitat for pupae [32]. Cylinders fit snugly into the lips of the pots, and rubber stoppers were used to seal the access holes. Depending on the size and branch number of each plant, 20–50 adult thrips were added to each cylinder through one of the access holes using a custom-built manual aspirator (aspirator, henceforth; Appendix A). Adult thrips that had emerged from pupae within roughly 7–10 d (recently emerged, henceforth) were selected from two to three different cages to maximize genetic diversity, and roughly equal numbers of males and females were chosen. Sex was estimated based on size, where females tend to be approximately 20% larger (Halbritter, unpublished data).

Cylinders were housed in a laboratory that had temperatures ranging from 23 to 24 °C and relative humidity ranging from 45 to 60%. The cylinders created a microclimate ranging from 22 to 25 °C and 52 to 88% RH, as measured by iButton Hydrochron Temperature/Humidity Loggers (DS1923; Maxim Integrated, Dallas, TX, USA) suspended in the center of a plant. Lighting was provided by four fluorescent grow bulbs (sunlite^®^ F54T4/865/HO; Sunlite, Brooklyn, NY, USA) held 22 cm above the tops of the cylinders. The photoperiod was set to 14:10 light:dark. Plants were watered two to three times per week by adding approximately 250 mL of tap water to plastic saucers in which each pot was resting. After most of the F_1_ larval thrips descended from the plants to pupate, plants were trimmed to roughly 15 cm above the moss. We had occasionally seen pupae in dried leaves and we, therefore, left the trimmed material resting on the moss. Emerging adults were collected using a passive trapping technique (Dyer, unpublished method): Cylinders were replaced with 3.8 L plastic jars (S-17077; Uline, Pleasant Prairie, WI, USA) in which the bottoms were cut out and inverted funnels (FSPP75; The Lab Depot, Dawsonville, GA, USA) were glued to the lids with their centers cut out (Figure 2B). Two 7 × 7 cm windows were cut and covered in mesh (250 µm or less) for ventilation. Floral tubes with silicone caps (Aquatube cat # 54–47; Syndicate Sales, Inc, Kokomo, IN, USA) were inserted onto the funnel spouts to collect emerging adult thrips that traveled up and toward the light. To minimize thrips death from starvation, a small leafy cutting of *Schinus* was added to each floral tube as soon as adult thrips began to emerge. Adults were collected every 1–3 d until thrips emergence stopped, and then the plants were trimmed and brought outdoors for regrowth and reuse. From cylinders and the other two cage types, efforts to collect adults commenced in the afternoon to maximize accumulation time during active daylight hours. Cylinders, plastic saucers, and jars were hand washed with tap water and dish soap after every use.

Given the amount of maintenance involved for each cylinder setup, we began to explore larger cages to see if we could increase the number of thrips produced with less effort. Rectangular 81.5 × 39.5 × 39.5 cm acrylic cages were custom ordered such that they had screen backs (thrips-proof nylon screen, 250 µm mesh), two 20 cm diameter access ports on the sides, 30 × 30 cm doors on the front, and one 20 cm diameter screen ventilation port above each door (Suncoast Plastic Fabrication, Inc, Valrico, FL, USA) (Figure 3). The tops and fronts were made up of clear acrylic (6.35 mm thickness), and the sides and bottoms were white acrylic (6.35 mm thickness). These and the cylinder constructions were meant to contain thrips while allowing light penetration, air circulation, and visual observation of the thrips life stages while minimizing opening the cages. We covered the two access ports with plastic to retain moisture because the back mesh provided adequate ventilation. The acrylic boxes were housed in the same lab as the cylinders and produced microclimates ranging from 23.5 to 25.5 °C and 50 to 77% RH. The same lights and photoperiod were provided, except lights were 17 cm above the tops of the boxes. Each box held two optimum-quality 3.8 L plants. Given the greater cage volume, less leaf trimming was needed, and plants taller than 45 cm were often used. Moss was added to pots and pots were placed in plastic saucers. Plants were watered in the same manner as cylinders. In total, 100 recently emerged adult thrips, roughly equal numbers of each sex, were added via aspirator to each box. We noticed that F_1_ larvae seeking refuge for pupation wandered out of the pots or possibly dropped off the plants and onto the bottoms of the boxes. Stacks of paper towels shaded by a piece of aluminum foil laid on top were provided as harborage on the cage bottoms. This method proved to be an effective way to collect pupae for use in other experiments. Once plants were largely void of larvae, all damaged stems and foliage were trimmed and laid on the cage bottoms. A few live leaves were left on the plants to serve as aggregation foci for emerging adults. Leaves with aggregations of newly emerged adults were collected from the boxes. After collection, plants were brought outdoors for regrowth and later use. Boxes were wiped with soapy sponges, then wet paper towels, and lastly dried with paper towels.

With the greater anticipated demand for thrips, we continued to investigate larger-scale production methods. We purchased 1.8 × 1.8 × 1.8 m walk-in style cages made from 280 µm Lumite^®^ screen (BioQuip^®^ Products, Compton, CA, USA) (Figure 4). The cages were a custom order with screen bottoms. To prevent wandering F_1_ thrips larvae from escaping, all corner seams were sealed with silicone. A zipper down the front provided walk-in access and PVC (19 mm diameter; ¾ inch; schedule 40) support frames were assembled from inside the cages. Black landscaping fabric was used to line the bottoms to protect the screen. Six 11.4 L plants, each with 20–40 stem tips, and each having 15–20 cm of tender new growth were added to each cage, and each plant was placed in its own plastic saucer. The average height, canopy diameter, and basal trunk diameters of plants used were 72.7, 76, and 2.4 cm, respectively. Plants with saucers were placed onto flat drip trays (54 × 42 × 7 cm) (cat # CN-FLHD; Greenhouse Megastore, Danville, IL, USA) with drainage holes, and drip trays were covered in a 3–5 cm layer of sphagnum moss. Additional moss trays (54 × 28 × 7 cm) were placed on the cage bottoms while leaving a narrow footpath to access the plants. Moss was also added to the growth medium surface of pots themselves up to just below the tops of the pots. Newly emerged adult thrips from earlier cages were collected via aspirator into six plastic vials and a total of 1500 thrips were added to start each screen cage. Vials were opened and left wedged in plant branches to allow the thrips to disperse onto the plants.

Screen cages were placed outdoors in the covered screen house. The floor was fine gravel covered with black landscape fabric. Microclimates within the cages reflected outdoor conditions (see results). Plants were watered three times a week from below by spraying tap water into the saucers to fill each to roughly 5 cm. The moss in the drip trays was also sprayed to help maintain moisture. When most of the F_1_ larvae had descended to pupate, dead plant material was routinely trimmed and scattered on the floor of the cages. Cages were monitored daily until the first F_1_ adults began to emerge and wander the walls and ceilings of the cages. Some of the emerging F_1_ adults would aggregate on the trimmed 11.4 L plants where live material remained and that material was routinely harvested. Most thrips were harvested using another passive trapping technique we developed, in which additional fresh plant material was provided to encourage aggregation behavior and facilitate the efficient harvest of the newly emerged adults (Figure 5). At any given time during the harvest interval, one to three 1.5 L plants were hung from the PVC frame along the ceiling nearest to where the greatest densities of wandering thrips were observed. The foliage of the plants was positioned such that it was in contact with both the wall and ceiling facilitating thrips movement. New plants were added to replace infested ones removed for harvest and, over the course of roughly 1.5 wk, six to eight hanging plants were harvested. The aggregated thrips seldom took flight when we handled the plants for removal, but we nonetheless transported plants to the lab in acrylic boxes as a precaution and for temporary storage before counting and preparation for field release. Counting and collection from trap plants were achieved via manual aspiration. Some thrips were reserved to reset cages. Trimmed 11.4 L plants were brought outdoors for regrowth. In 36% of screen cage resets, moss was dried in an oven at 70 °C for at least 3 d before being reused. Cages were swept or vacuumed clean before being reset.

### 2.3. Statistical Analyses

Screen cage microclimates were measured from September 2020 to December 2020. In that time, cages experienced the full range of outdoor temperatures and humidity that occurred in Davie, FL, from December 2019 through June 2021 (Florida Automated Weather Network; https://fawn.ifas.ufl.edu, accessed on 17 June 2021). As cage temperature and humidity were not recorded for the entire time period, we used the equation produced from a linear regression between cage microclimate and weather station data from September 2020 to December 2020 to extrapolate the missing microclimate data. We obtained hourly temperature and humidity data for the remaining dates between December 2019 and April 2021 by inputting weather station data into the equation to yield the microclimate estimates. To investigate the correlation between microclimate and fold productivity (the number of F_1_ thrips harvested divided by the number of F_0_ thrips introduced) and generation time (time from F_0_ adults introduced to first F_1_ thrips harvested), we created temperature and humidity profiles for each screen cage replicate (*n* = 54) that spanned the time between when each cage was first set up and the date the last F_1_ thrips were harvested. Within these time range profiles, we determined the mean temperature and humidity. Fold productivity and generation times were modeled as functions of temperature and humidity to examine correlations. A Kruskal–Wallis test was used to test for a difference in fold productivity and generation time among the three cage types and post hoc comparisons were made using the Dunn test. All statistical tests and models were conducted in R version 3.3.3 [33].

## 3. Results

### 3.1. Comparisons between Cage Types

Screen cages facilitated the shortest average generation time of 32.9 days (W_2_ = 62.861, *p* < 0.001). Acrylic cylinders and screen cages facilitated the greatest fold productivity, while acrylic boxes were the least productive (W_2_ = 13.425, *p* = 0.001). When considering the number of personnel hours needed to rear 1000 thrips, screen cages were an order of magnitude more efficient than either of the two acrylic cages, requiring roughly ½ h per 1000 thrips (Table 1).

### 3.2. Temporal Trends in Production

Rearing data were collected from cylinders from 26 August 2019 to 3 June 2021, during which 69,503 thrips were produced. Rearing data were collected from acrylic boxes from 18 September 2019 to 7 July 2021, during which 61,768 thrips were produced. For either enclosure, the number of thrips produced, and generation times tended to vary considerably with no clear or consistent seasonal patterns (Figure 6).

Rearing data were collected from screen cages from 20 December 2019 to 17 June 2021, during which 748,633 thrips were produced. Production and generation time tended to vary seasonally, with increases in production in the late spring/early summer and late summer/early fall (Figure 7A) and shorter generation times in the summer (Figure 7B).

### 3.3. Temperature and Humidity Variations in Screen Cages

Average monthly temperatures in screen cages varied seasonally from 18.9 to 28.5 °C (Figure 7). At the hourly scale, extremes in screen cage microclimates ranged from 4.6 to 34.3 °C and 54.8 to 99.8% RH. There was a significant, albeit weak, positive correlation between mean temperature and the number of thrips per batch in screen cages (Figure 8A). There was a slightly stronger, negative correlation between mean temperature and generation time in screen cages (Figure 8B). Mean humidity did not correlate significantly with thrips production in screen cages.

## 4. Discussion

Our work toward developing a mass rearing technique that yields prolific numbers of *P. ichini* builds upon that of others, from which a method to rear *P. ichini* for quarantine colonies and experiments was initially developed. We formally quantified the production efficiency of the original acrylic cylinder technique, both with respect to thrips fold productivity and personnel hours required. The average thrips productivity of 368 per generation in our cylinders was comparable to that from others in specific experimental studies. Wheeler et al. [25] report thrips production per cylinder averaging roughly 220 per generation, and Manrique et al. [21] report roughly 160 larvae produced per cylinder per generation. Cuda et al. [26] report up to 3153 *P. ichini*/*P. gandolfoi* produced per month using cylinders for part of colony production. Our average total monthly production for cylinders was comparable at 3310 thrips from a monthly average of nine cylinders. With an average of three screen cages per month, we were able to produce a monthly average of 44,037 thrips with roughly 1/10 of the personnel hours required for the cylinder equivalent, highlighting the marked advantage of using larger cages to rear *P. ichini*.

Being small-bodied poikilotherms, insects are strongly affected by ambient temperatures. Not surprisingly, the development time for *Pseudophilothrips* spp. varies considerably with temperature [21,34]. Generation time appeared to decrease as temperatures peaked during the summer months in the outdoor screen cages as would be expected, but the correlation was weak. Additionally, thrips productivity appeared to increase with temperature for screen cages, but again, it was only weakly correlated and a drop in productivity was observed in mid-summer. We analyzed mean temperatures over the course of the entire generation for each batch, but it has been shown that temporary and/or repeated exposures to extreme temperatures can impact fitness in other insects [35,36]. This remains for a future study in which the impacts of extremes and their duration of exposure will be quantitatively studied for *P. ichini*. Laboratory conditions were stable in our study at roughly 23.5 °C, yet we saw considerable variations in both generation time and fold productivity throughout the year for thrips reared in acrylic cylinders and boxes. The random variability of these indoor productivity metrics may be due to variations in individual plant quality and/or changes in plant quality associated with seasonal transitions in plant phenology. Therefore, the temperature may still impact indoor production but in an indirect way.

Host plant quality is a crucial component impacting the fecundity of phytophagous insects [37]. For example, plant quality associated with variations in fertilization can impact aspects of the life history of *P. ichini* [30], as well as fecundity [25]. However, our plants were all fertilized at the same rate. Seasonal changes in plant phenology can impact the allocation of resources to specific plant tissues [38]. *Schinus* flowering peaks in October, and fruit ripens from December through March; most plants are flushing with vegetive growth by early spring [39]. The latter period appears to translate to increases in thrips fold productivity in screen cages later in spring and early summer. Some plants will flower again in April [39], which potentially contributed to the downstream drop in production in mid-summer despite the warmer temperatures. Our plants were trimmed to continuously produce new flushes throughout the year, but flowering did occur at low densities on some plants. Quantification of foliar nitrogen content and defensive compounds at monthly intervals could provide insight into seasonal changes in plant quality potentially impacting the production of *P. ichini*. The signal of seasonal variations in thrips productivity was apparent in the outdoor screen cages, but variation appeared random throughout the year for the cylinders and boxes. Although the latter two cage types were housed indoors, plants were still running their phenological clocks from their time growing outdoors prior to being brought indoors. It may be because, given each batch of indoor thrips was reared on one or two plants, individual variations in plant quality resulted in noise that masked a seasonal signal, while in screen cages, the signal was more apparent because each batch was reared on six plants and plant variations “averaged out” for each batch of thrips.

Internal water balance is another important factor for insect survival owing to their small size and large surface-area-to-volume ratios. Although we did not find a significant correlation between humidity and any production metric, it is possible that extremely low humidity can impact the non-feeding stages of *P. ichini*. Pupal survival to the adult stage did not differ significantly between *P. ichini* pupae held at 50% RH versus 100% RH, but evidence suggested adults from the 100% RH groups were more robust, as more were able to escape the experimental arenas (Halbritter, unpublished data). The moss harborage we provided in screen cages was routinely misted, and moss microclimates would not have likely dropped below the ambient minimum of 54.8% we recorded. Nonetheless, future studies on the impact of extremes in humidity would benefit our knowledge of mass rearing small, soft-bodied phytophagous insects such as *P. ichini.*

## 5. Conclusions

The screen cage method we optimized for mass rearing *P. ichini* allowed us to keep up with stakeholder demand for field release across the invaded range of *Schinus* in southern Florida—between 24 May 2019 and 2 July 2021, our facility had released 962,896 thrips in 14 counties. The passive trapping method developed here allows the efficient harvest of thrips from the screen cages, minimizing the time spent working inside the cages when working conditions are extremely uncomfortable during the summer months. The greatest numbers of thrips harvested from the outdoor screen cages occurred April through June, with a smaller peak in thrips harvested August through September. Acrylic cylinders and boxes are useful for maintaining small colonies for experiments and allow convenient observation of thrips behavior and development in a comfortable laboratory environment. When upfront costs of the cages versus thrips outputs are considered, cylinders are the best option for small laboratory operations, as they are more cost efficient than acrylic boxes. While most expensive, screen cages are at least four times more cost efficient than either of the two small cage types. Our work highlights the importance of understanding agent biology and stage-specific requirements in adapting existing techniques to meet the demand for beneficial insects used as agents in classical or augmentative biological control. Our methods have transferability to other programs where intact host plants are needed to rear small insects such as thrips.

## Figures and Tables

**Figure 1 insects-12-00790-f001:**
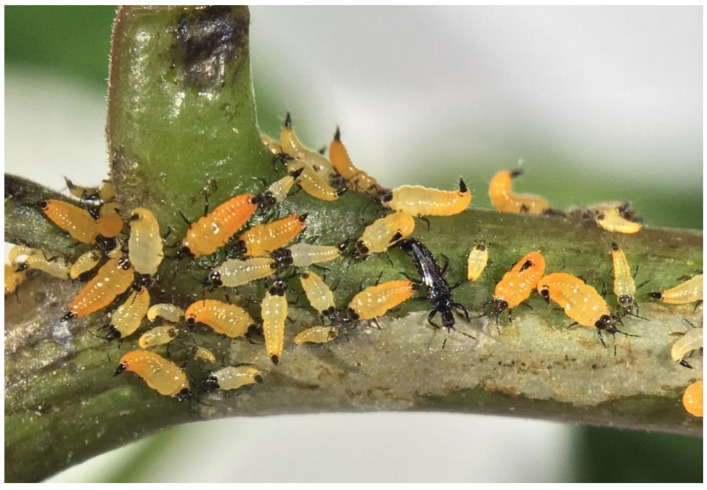
A feeding aggregation of thrips on a succulent, flushing Brazilian peppertree stem a few cm away from the heavily damaged apex. First and second instar larvae are bright orange with black heads, antennae, pronota, legs, and tips of the abdomen, and the winged adult is black.

**Figure 2 insects-12-00790-f002:**
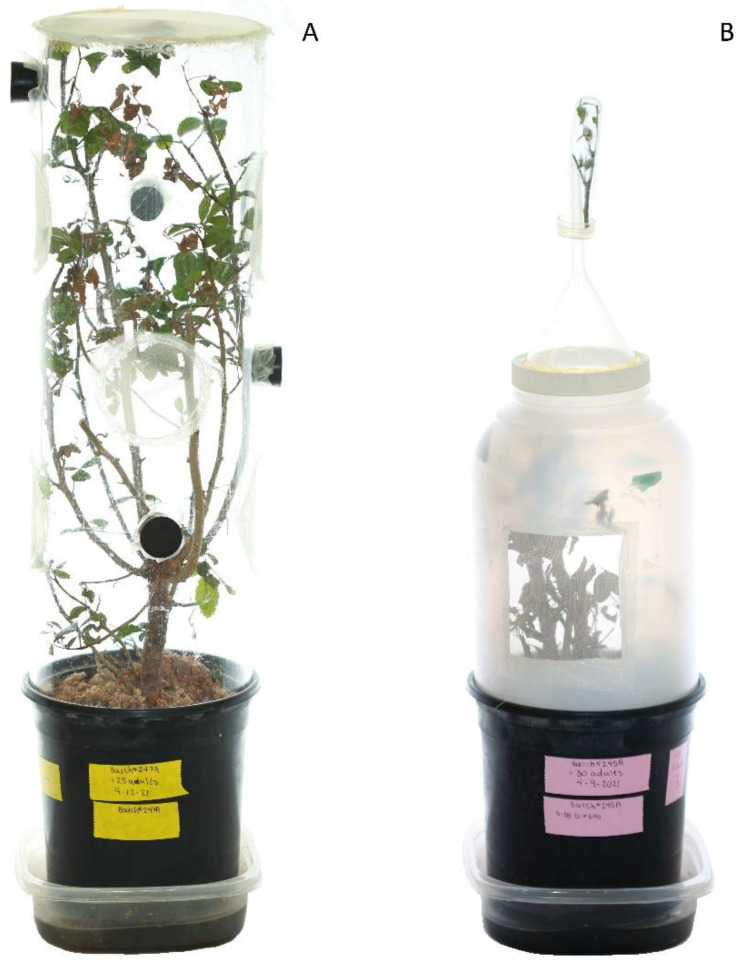
(**A**) Vented acrylic cylinder placed over a 3.8 L Brazilian peppertree sapling. The cylinder was 45 cm tall and 15.5 cm in diameter. The open top and six 7 cm diameter ventilation holes were covered in a 250 µm screen. Rubber stoppers were used to seal 2.5 cm diameter access holes; (**B**) passive collection apparatus for harvesting newly emerged adult thrips from acrylic cylinder cages. Adult thrips walked up the 3.8 L opaque jar into an inverted funnel and then accumulated in a small tube into which the spout of the funnel was inserted. The tube contained a leafy cutting of fresh Brazilian peppertree for food.

**Figure 3 insects-12-00790-f003:**
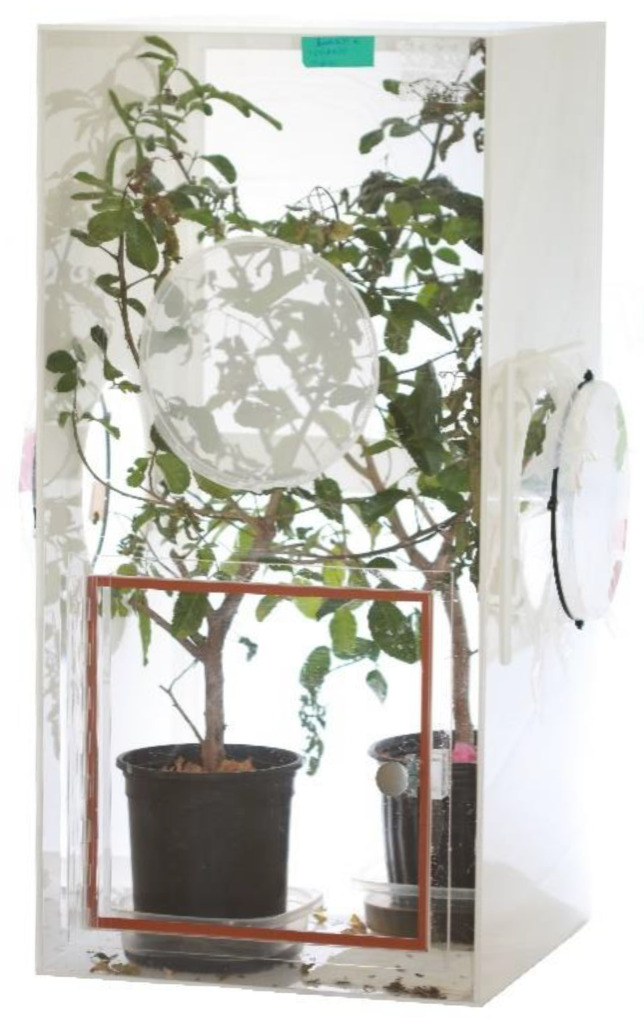
Acrylic box (81.5 × 39.5 × 39.5 cm) used to house two 3.8 L Brazilian peppertree saplings. Access was via a hinged door at the bottom of the front panel, and the back and 20 cm diameter opening on the front were covered in a 250 µm screen for ventilation. The 20 cm diameter side openings were covered in plastic.

**Figure 4 insects-12-00790-f004:**
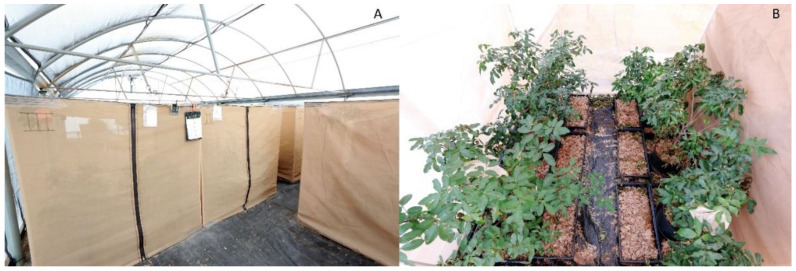
(**A**) Several large, 1.8 × 1.8 × 1.8 m, 280 µm mesh screen cages used for mass production in an outdoor, covered screen house; (**B**) inside view of one of the large screen cages. Black landscape fabric lined the floor and moss-filled drip trays provided pupation substrate. Six 11.4 L Brazilian peppertree saplings were placed in each cage.

**Figure 5 insects-12-00790-f005:**
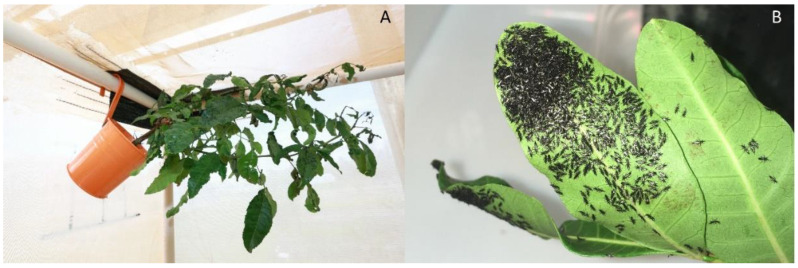
A passive trapping method to harvest emerging adult thrips from large, 1.8 × 1.8 × 1.8 m screen cages: (**A**) a small, 1.5 L Brazilian peppertree sapling was hung from the frame of the screen cage and emerging adult thrips initially traveling up and toward the light encountered the fresh plant material and began to aggregate; (**B**) dense aggregation of adult thrips on leaves.

**Figure 6 insects-12-00790-f006:**
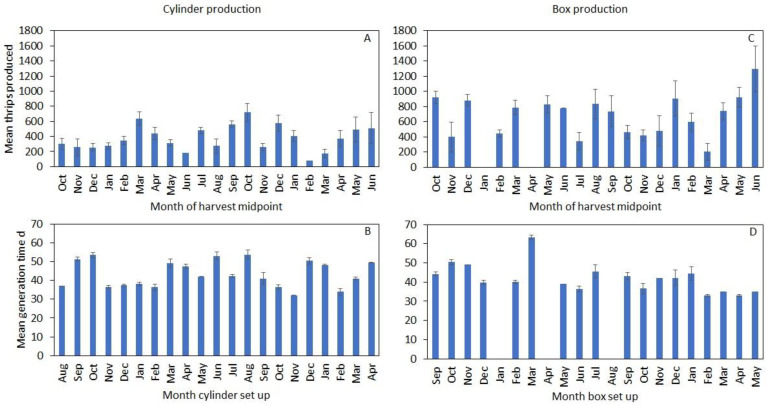
Monthly acrylic cylinder and box production metrics, where the means for thrips produced were grouped by month based on the midpoint of thrips harvest, and the means for generation time were grouped based on the month cylinders/boxes were set up. The series for cylinders begins in August 2019 and that for boxes begins in September 2019: (**A**,**C**) mean monthly thrips production; (**B**,**D**) mean monthly generation times, where generation time refers to the time between the setup of a cylinder/box and the first harvest of F_1_ adults. No boxes were set up in the months of January, April, or August 2020.

**Figure 7 insects-12-00790-f007:**
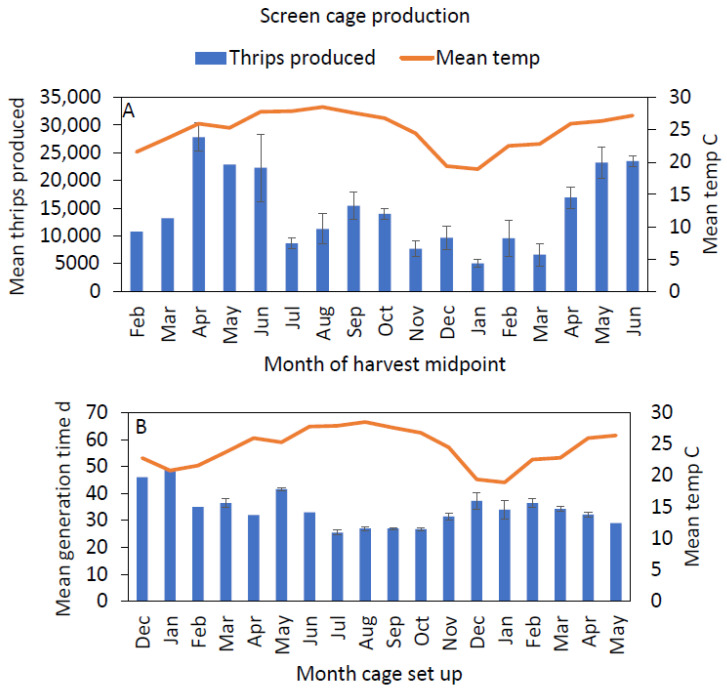
Monthly screen cage production metrics, where the means for thrips produced were grouped by month based on the midpoint of thrips harvest, and the means for generation time were grouped based on the month screen cages were set up. The series begins in December 2019: (**A**) mean monthly thrips production; (**B**) mean monthly generation times, where generation time refers to the time between the setup of a caged screen and the first harvest of F_1_ adults. The orange curve represents the mean temperature within the cages for each month.

**Figure 8 insects-12-00790-f008:**
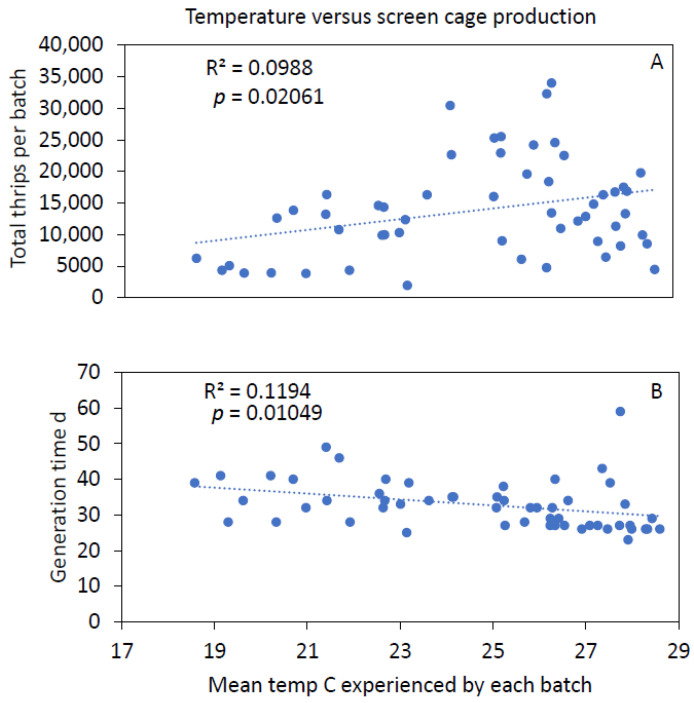
Correlation between the mean screen cage temperature for the entire generation time each of 54 batches and the number of thrips produced in each respective batch (**A**) and the generation time of each respective batch (**B**).

**Table 1 insects-12-00790-t001:** Comparative thrips productivity among the three different cage types used for rearing. Harvest methods used for cylinders, boxes, and screen cages were as follows: a jar covering with inverted funnel ending in a floral tube, removal of remnant leaves with thrips aggregations, and removal of small hanging plants with thrips aggregations, respectively. A batch (i.e., cohort) refers to one generation of thrips, where the batch was harvested as the F_1_ thrips reached the adult stage. Fold productivity, number of F_1_ thrips harvested, and generation time are presented as mean ± SE per batch. Fold productivity and generation time were statistically tested for differences with respect to cage type. Hours of labor are reported as the time from cage setup to final breakdown and normalized per 1000 thrips. Labor breakdown by task can be found in the Appendix A. Upfront cost per cage (US dollar) is only for the cage itself but the cost for cylinders also includes the jar, funnel, and floral tube. Year refers to the most recent year items were purchased.

Cage Type	Number of Batches	Number of F_0_ Thrips Introduced	Number of F_1_ Thrips Harvested	Fold Productivity ***	Generation Time (d) ***	Hours of Labor/1000 Thrips	Upfront Cost/Cage (Year)
Acrylic cylinder	189	20–50	368 ± 20	10.4 ^a^ ± 0.6	42.5 ^a^ ± 0.6	5.0	USD 62 (2019)
Acrylic box	91	100	679 ± 38	6.8 ^b^ ± 0.4	43.3 ^a^ ± 0.9	5.1	USD 400 (2014)
Lumite^®^ screen	54	1500	13,864 ± 1044	9.2 ^a^ ± 0.7	32.9 ^b^ ± 0.9	0.5	USD 600 (2020)

*** *p* < 0.001. Different letters indicate differences in group means based on post hoc comparisons.

## Data Availability

The data summarized in this study are available in the Appendix A.

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
