# Peer review of "Advances in Mass Rearing Pseudophilothrips ichini (Hood) (Thysanoptera: Phlaeothripidae), a Biological Control Agent for Brazilian Peppertree in Florida"

_insects, 2021, doi:10.3390/insects12090790_

Round 1

Reviewer 1 Report

General: Good manuscript, provides complete information on rearing methods for this small and somewhat difficult to manipulate biological control agent. The manuscript demonstrates a successful scale-up from quarantine lab rearing for research studies to mass-rearing at a level sufficient to release over 960K thrips for biocontrol of Brazilian peppertree in a short time-just since the summer or fall of 2019. success if defined by achieving the same or better 9 to 10 fold-production output with 10-fold less labor in the final mass-rearing protocol. The protocol navigated the need to provide both food and pupation substrate.The authors should specify what their labor calculation includes. The protocols will likely be of use for rearing of other small insects that feed on liquid cell contents. Only minor comments.

Abstract:

Suggest mentioning that the outdoor cage method achieved the same 9 to 10-fold fold output efficiency as lab cylinder rearing, but with 10-fold less labor per thrip.

Introduction

Line 68-69: Are the adults strong fliers? Or do they disperse more by crawling? (relevant for ease of collecting adults from ‘hanging’ plants in the Lumite cage method).

Line 90-92: Delete “otherwise” and start sentence with “Details…”. Also, what are the “…other studies…” that assessed cylinder productivity? Please cite.

Line 102-105: Delete sentence that starts “Our method…” (not needed for Introduction).

Materials and Methods

L109: Clarify what “drupes” means for non-botanists (fruits).

Plant production: Question on the need to move Brazilian peppertree plants up to four times (from germination trays to 1.5L pots to 3.8L pots to 11.4 L pots). Was any effort made to streamline plant rearing (reduce # of transfers)?

L184: Figures provided in the article body should start with Figure 1, not 2. The supplemental Figure is numbered Figure S1.

L202: Please indicate and reference expected size ranges of female and male adults.

L231: Were thrips collected from the 3.8L-pot-based emergence device at a specific time of day (ie late in the afternoon to maximize light-trapping)?

L310: Six to eight hanging plants were used to collect adults from each Lumite cage? Also, were the hanging plants collected at a specific time of day to maximize emergence? Also, how were these hanging plants contained/processed after removal from the Lumite cage? How were the thrips collected from the ‘trap’ plants? How were they stored and transported for field release?

General: How were labor-hours per thrip determined? What was included in labor?

Statistical Analyses: It is not clear how temperature was extrapolated from the four month period of Sept-Dec 2020 to the entire rearing period of Dec 2019 to April 2021. Why not just use climatic data from the nearest weather station for the entire period, to examine the effect of temp and RH on the Lumite cage production?

Results

Table 1: Please clarify that the output thrips harvesting method used to obtain data for output counts was the inverted jar/funnel/florist water pik method for the cylinder rearing method; the remnant leaf aggregation method for the acrylic rectangular cage method; and aggregation of adults on leaves hung from the ceiling for the Lumite cage method.

Discussion

L453 vs 382. Differing values of minimum RH recorded in the Lumite cages (55.5 vs 54.8%, respectively). Unless the former refers to weather station data and the latter to loggers inside cages.

Literature Cited

20, Manrique (not “anrique”).

Reviewer 2 Report

This manuscript describes the development of a method of passive capture for efficient harvesting of thrips from screen cages and large-scale production of Pseudophilothrips ichini.

In general, the article is very interesting. The study design and
methods are clear enough and the provided illustrations helped understanding what was done; procedures are sufficiently described. Data is clearly presented.
The manuscript is rich in methodological details, considering its purpose.

Regarding the description of methods, i suggest reducing information in some points. I did not point out which paragraphs should be changed. I leave it for the authors to decide.  

The authors show the number of staff hours needed to
create screen cage thrips, and my question is: How much does it cost?
Even if it is not the objective of the study, the economic issue is
very important, considering that, according to the authors, the method could
be transferred to other studies where intact host plants are needed to raise small insects.

Item 2.2, Were the breeding cages installed in Davie, Florida, in the same place where the fruit collection took place?
